# *Staphylococcus aureus* Causes the Arrest of Neutrophils in the Bloodstream in a Septicemia Model

**DOI:** 10.3390/microorganisms10091696

**Published:** 2022-08-24

**Authors:** Svetlana N. Pleskova, Sergey Z. Bobyk, Ruslan N. Kriukov, Ekaterina N. Gorshkova, Nikolay A. Bezrukov

**Affiliations:** 1Laboratory of Scanning Probe Microscopy, Lobachevsky State University of Nizhny Novgorod, Gagarina Ave., 23, 603950 Nizhny Novgorod, Russia; 2Department “Nanotechnology and Biotechnology”, R.E. Alekseev Technical State University of Nizhny Novgorod, Minina St., 24, 603155 Nizhny Novgorod, Russia; 3Department of Semiconductors Physics, Electronics and Nanoelectronics, Lobachevsky State University of Nizhny Novgorod, Gagarina Ave., 23, 603950 Nizhny Novgorod, Russia; 4Department of Molecular Biology and Immunology, Lobachevsky State University of Nizhny Novgorod, Gagarina Ave., 23, 603950 Nizhny Novgorod, Russia

**Keywords:** *Staphylococcus aureus*, neutrophils, endotheliocyte, force of adhesion, work of adhesion, receptors, NETs

## Abstract

*Staphylococcus aureus* induces the expression of VCAM-1, P- and E-selectins on the endothelial cells of the EA.hy926 cell line but, at the same time, causes the significant suppression of the force and work of adhesion between these receptors of endotheliocytes and the receptors of neutrophils in an experimental septicemia model. Adhesion contacts between the receptors of neutrophils and endotheliocytes are statistically significantly suppressed under non-opsonized and opsonized *S. aureus* treatment, which disrupts the initial stage of transendothelial migration of neutrophils—adhesion. Thus, *S. aureus* causes the arrest of neutrophils in the bloodstream in an experimental septicemia model.

## 1. Introduction

*Staphylococcus aureus* is one of the most common causes of morbidity and mortality from infections worldwide. This pathogen can cause a wide range of illnesses, including serious pneumonia and fatal sepsis [1]. Sepsis is characterized by significant mortality rates: Even with proper treatment, its mortality reaches 50%, depending on the severity of the infection. In addition, sepsis is characterized by frequent relapses (5–10%) and various complications in more than one-third of its survivors [2,3]. In recent years, the incidence of *S. aureus* bloodstream infections has been on the rise in developed countries [4]. The key *S. aureus* virulence strategies in their pathogenesis are the secretion of coagulases, which bind and activate prothrombin, and the exposure of bacterial surface agglutinins, which bind polymerized fibrin. The culmination of these processes is the formation of abscesses [5]. Thus, active immunosuppression, often combined with inflammation, can be a typical characteristic of sepsis. The endothelium of blood vessels is actively involved in the development of sepsis since proinflammatory and procoagulant factors are secreted on its surface. At the same time, uncontrolled active inflammation causes damage to the endothelium itself, leading to vascular destruction and an inability to maintain normal blood pressure [4,6]. The most important factors of non-specific defense are neutrophils (in the blood and tissues) and macrophages (in tissues). However, ingested *S. aureus* can persist inside phagocytes like “Trojan horses” and spread throughout the body, causing systemic dissemination [7,8]. *S. aureus* deftly uses the primary immune response for its purposes, also interacting with the coagulation system or endothelium, and can regulate the expression of virulence factors in the bloodstream [4]. To be activated (primed), neutrophils must move to the marginal pool of the bloodstream and adhere to endotheliocytes; in addition, the adhesion stage is fundamentally important for subsequent extravasation. During the next step, rolling, the binding of E- and P-selectins of the endothelium with the corresponding carbohydrate ligands of neutrophils, such as P-selectin glycoprotein ligand (PSGL-1) and E-selectin ligand-1 (ESL-1), occurs [9]. In this case, the expression of endothelial selectins is strictly regulated, which is necessary to avoid autoimmune reactions: P-selectin from Weibel–Palade bodies and E-selectin synthesized de novo can move to the apical membrane of endothelial cells, changing the expression density and interaction intensity [10]. The nature of the neutrophil interactions with various selectins is also different: It is supposed that P-selectin provides only the initial “recognition” of cells, while E-selectin causes a more stable slow rolling [11]. *S. aureus* can influence the interaction between neutrophils and endotheliocytes. It has been noted that particular strains of *S. aureus* are able to induce the expression of selectins (E-selectin) and integrins (ICAM-1) [12]; in addition, a high level of E-selectin expression correlates with the severity and duration of bacteremia in patients infected with *S. aureus* [13]. *S. aureus* also affects other participants in the adhesion process, namely neutrophil granulocytes, with a special toxin staphylococcal superantigen-like 11 (SSL11), which interacts with the sialyl Lewis X (CD15) of neutrophil. It induces the arrest of phagocytes on endotheliocytes without the induction of oxidative burst [14]. Therefore, the study of adhesion as an initial stage of interaction between neutrophils and endotheliocytes, as well as the influence of *S. aureus* on this process, is a fundamental issue.

A unique approach for the study of submicron structures, atomic force microscopy (AFM), made it possible not only to perform microscopic studies but also to study the viscoelastic properties of the living cell in a native environment [15]. In 2005, Eibl and Moy developed a fundamentally new method for AFM application, according to which one cell was attached to the cantilever, and the other adhered to the substrate. Thus, adhesive contacts were formed between the cells after the cantilever approached the substrate, and the force of these contacts could be calculated after the cantilever retreated [16]. In practice, it was the modified regime of AFM-based FS spectroscopy. In one study [17], the bonds of lymphocyte function-associated antigen 1 (LFA-1 or CD11a/CD18) and macrophage-1 antigen (Mac-1 or CD11b/CD18) with the respective ligands were studied using a modification of the FS spectroscopy method: single-molecular AFM. This method was used not only for the measurement of cell-to-cell interaction affinity but also for the estimation of its lifetime. Later, some of these methods were tested in clinical practice. For example, Guedes et al. (2019) [18] used single-molecular AFM to show the difference in the interaction between fibrinogens and erythrocytes among healthy donors and patients with essential arterial hypertension, indicating that the patients had higher levels of these factors. In another study [19], the Eibl and Moy system was significantly expanded, which allowed the estimation of the work of adhesion in addition to force measurement. Using this system, the total force of interaction between two neutrophils and two lymphocytes from one donor was obtained. Additionally, it was determined that the measured adhesion force between neutrophils was higher than that between lymphocytes.

The tasks of this work were to (1) study the changes in the expression profiles of the endothelial cell membranes after *S. aureus* infection using flow cytometry; (2) measure the force and work of adhesion between the selectins expressed by endothelial cells and their ligands on the neutrophil surface using FS spectroscopy; (3) assess the conditioning effect of *S. aureus* on this interaction; (4) estimate the influence of *S. aureus* opsonization on this process; and (5) assess the changes in neutrophil morphology and viability after transendothelial migration using fluorescence microscopy.

## 2. Materials and Methods

### 2.1. Preparation of Human Neutrophils

Blood from 6 healthy volunteers (male and female) between 20 and 40 years old was provided by the Nizhny Novgorod N.Y. Klimova Regional Blood Center. The study was approved by the Commission on Bioethics of the N.I. Lobachevsky State University of Nizhny Novgorod (created on 11 November 2016, order for the creation No. 497-OD), protocol No. 9 dated 17 July 2017. To prevent the formation of clots, blood was incubated in PBS with heparin (25 U/mL) in equal proportion. Neutrophils were isolated via centrifugation using two ficoll–trasograph solutions (ρ = 1.077 g/mL, ρ = 1.117 g/mL, 420 g, 40 min) [20]. Then, neutrophils were washed with PBS twice (420 g, 3 min) and used at a concentration of 1 × 10^6^ cells/mL. The viability of the neutrophils used in the experiment, confirmed by staining with propidium iodide, was at least 99%. Siliconized tubes were used to prevent neutrophil priming.

### 2.2. Cell Culture

EA.hy926 human endothelial cells (ATCC, CRL-2922) were cultured in 25 cm^2^ flasks (Corning, Somerville, MA, USA) in complete DMEM/F12 medium (PanEco, Moscow Russia) supplemented with 10% fetal bovine serum (PanEco, Russia), 100 U/mL penicillin, 100 μg/mL streptomycin (PanEco, Russia), 8 mM glutamine (PanEco, Russia), HAT (100 μM hypoxanthine, 0.4 μM aminopterin, and 16 μM thymidine) (Sigma-Aldrich, St. Louis, MO, USA) in a CO2 incubator (5% CO_2_) in a humid environment (Binder, Neckarsulm, Germany) [21]. For the AFM study, cells were seeded on Petri dishes (35 mm, Corning^®^ Treated Culture Dishes, Somerville, MA, USA), with 4 × 10^5^ cells per Petri dish, cultivated under standard conditions for two days, washed with Hanks’ buffered solution (HBS) (2 mM L-glutamine and 10 mM HEPES). The end cell’s concentration was approximately 1.4 × 10^6^ cells per Petri dish.

### 2.3. S. aureus Culturing

The cultivation of the *S. aureus* 2879 M strain was described in [22]. Briefly, *S. aureus* 2879 M was cultured (37 °C, 24 h) in meat-peptone agar; cells were washed from the medium via centrifugation 3 times with sterile PBS (1800 g, 10 min), resuspended in PBS, and then adjusted to the optical density of 0.75 (λ = 670 nm), which corresponded to 1 × 10^9^ cells/mL, using a colorimeter (KFK-2MPUHL, Russia). The bacterial suspension was incubated with pooled sera from three donors (60 min, 37 °C, shaking every 10 min) and then washed three times with PBS for the experiment with opsonized bacteria. To assess the cells’ adhesion, both opsonized and non-opsonized bacteria were used in a final concentration of 5 × 10^7^ cells/mL (approximately 140 bacteria per 1 endotheliocyte). For migration experiments, the concentration was 2.5 × 10^7^ cells/mL.

### 2.4. The Force and Work of Adhesion Determination

The determination of the force and work of adhesion was described in [23]. An NTegra atomic force microscope (NT-MDT, Moscow, Russia) mounted on an inverted light microscope (Olympus, Tokyo, Japan) was used for the AFM-based FS spectroscopy measurement of the force and work of adhesion between endothelial cells and neutrophils. A Si_3_N_4_ probe (C-MSCT, Bruker, Billerica, MA, USA) with f_0_4–10 kHz and k—0.010 N/m was used for attaching the neutrophils. For better attachment of the cells, the cantilever was coated with poly-L-lysine (Merck, Kenilworth, NJ, USA), then washed three times with a sterile normal saline solution (10 μL), and incubated (37 °C, 20 min) with cell suspension (10 μL). After that, the probe with neutrophils was washed three times with PBS. The concentration of neutrophils was selected in a series of preliminary experiments. The prepared cantilever with cells was put on a control head.

The probe with a neutrophil was placed above the endothelial cell monolayer under the control of the light microscope, and then the signal was registered in the contact mode. The distance between the cantilever and the cell varied from 100 to 5000 nm. The impact force was 10–50% of the maximum allowable, and the curve time ranged from 2 to 60 s. The maximum load force value was 3–4 nN. The approach offset time varied from 0.1 to 2.0 s, and 2.0 s was chosen as the optimal. The speed of tip movement was 5 μm/s. The work of Oberleithner et al. [24] and the series of our preliminary experiments were used for the main mechanic and kinetic parameter calculations. After obtaining the FS curves, feedback was temporarily disabled in order to change the probe area and avoid damage to the cell. Each sample was examined for 30 min; then, both the neutrophil on the tip and the endotheliocyte on the Petri dish were replaced. FS curves were calculated for all control measurements, and we did not observe any significant differences between the adhesion force and work data at the beginning and at the end of the experiment. Therefore, the affinity of the receptors was also not changed, and the 30 min time point was used for measuring the adhesive contacts. The force and work of adhesion were calculated by the following formulas:(1)$$\Delta F=F_{p}-F_{a}$$
where Δ*F* is the force required to break the bond, which corresponds to the difference in the value of the measured force before (*F_p_*) and after (*F_a_*) the rupture of the receptor bond.
(2)$$A=\overset{n}{\underset{i=1}{\sum }}F_{i}\times x_{i}$$
where *A* is the adhesion work, and *F_i_* and *x_i_* are the force and distance measured at the *i*-point of the FS curve.

After the estimation of the force and work of adhesion between the neutrophil and endothelial cell (control), a bacterial suspension of *S. aureus* 2879 M (opsonized or non-opsonized) was added for interaction conditioning. The magnitude of the maximum loading force, the approach–displacement time, the speed of movement, and the depth of the immersion of the tip into the cell were the same for the control (interaction between the cells without bacteria) and experiment (interaction between the cells with *S. aureus* 2879 M). For statistical analysis, 21 FS curves were assessed for each independent measurement with (experiment) or without (control) *S. aureus*. The blood neutrophils from 5 healthy donors were used to study non-opsonized *S. aureus*, and the blood neutrophils from 6 healthy donors were used to study opsonized *S. aureus*.

### 2.5. Adhesion Molecules’ Expression Estimated by Using Flow Cytometry

Using flow cytometry, the main selectin and integrin molecules (VCAM-1, ICAM-1, P- and E-selectins, and endoglin (ENG)) were estimated after the activation of the EA.hy926 culture by *S. aureus* 2879 M. For that, cells (5 × 10^5^) were seeded in a 24-well plate and grown (Corning Inc., Somerville, MA USA) in a CO_2_ incubator (1–4 h, 37 °C, 5% CO_2_) until the achievement of a continuous monolayer. Then, a suspension of *S. aureus* (2.5 × 10^7^ cells/mL) or tumor necrosis factor (TNF) in a concentration of 25 ng/mL (positive control) was added. After that, the cells were taken off from the culture plate using 0.25% trypsin EDTA (PanEco, Russia). Anti-endoglin antibodies (Cloud Clone Corp., Houston, TX, USA) (as secondary antibodies, FITC-labeled antibodies against mouse IgG (Sigma-Aldrich, St. Louis, MO, USA) were used), anti-CD62E (clone 1.2B6), specific to human E-selectin and P-selectin (Millipore, Burlington, MA, USA), anti-VCAM-1 (clone 429) labeled with phycoerythrin (PE) (eBioscience, San Diego, CA, USA), and anti-ICAM-1 (clone MEM-111) labeled with fluorescein isothiocyanate (FITC) (Invitrogen, Waltham, MA, USA) were used at a dilution of 1:1000 to measure the level of expression of the corresponding adhesion molecules on the surface of EA.hy926 cells in the native state and after exposure to *S. aureus* and TNF. The mean fluorescent intensity (MFI) values for each antibody were assessed with a Cytoflex S flow cytometer (Beckman Coulter, Brea, CA, USA) using the CytExpert Software v.1.2 (Beckman Coulter, Brea, CA, USA).

### 2.6. Transendothelial Migration of Neutrophils

The Ea.hy926 cells were cultured for 48 h under the conditions described in Section 2.2 until the formation of a monolayer in the upper compartment of a Transwell insert with a pore size of 0.4 µm (Corning, Somerville, MA, USA). The neutrophils were isolated as described in Section 2.1 and seeded with 150 µL of cell suspension (1 × 10^6^ cells/mL) in the upper compartment of a Transwell insert, and 600 µL of the *S. aureus* suspension (2.5 × 10^7^ cells/mL) in HBSS buffered by 10 mM HEPES was used as a chemoattractant (in a lower compartment). The HBSS medium buffered by 10 mM HEPES was used as a negative control. After incubation for 2 h at 37°C and 5% CO_2_, the cells in the lower compartment were fixed with 70% ethanol and stained using propidium iodide (PI) and FITC-labeled myeloperoxidase antibodies (clone MPO455-8E6) (Thermo Scientific, Waltham, MA, USA), diluted 1:50. The stained cells were counted using a Carl Zeiss Axio Vert A1 fluorescence microscope with a ×20 bright-field air objective and the Zeiss ZEN software (Carl Zeiss, Jena, Germany). Six independent experiments with the neutrophils of 6 different healthy donors were performed.

### 2.7. Statistics

The Origin 7.0 Server Package software was used to determine the mean values and standard deviations. The normal distribution was estimated by the Mann–Whitney test. The significance of differences between two values (control data and experimental values) was assessed by Student’s *t*-test in normal distributions and the Wilcoxon test in non-parametric distributions.

## 3. Results and Discussion

### 3.1. Conditioning Effect of S. aureus 2879 M on the Endothelial Cell Expression Profile

Changes in the expression levels of endothelial cells’ adhesion molecules (EA.hy926) after incubation with *S. aureus* can be seen in Figure 1. The classes of adhesion molecules were chosen based on their most frequent expression on the surface of the EA.hy926 cell line [25]. The graphs show that *S. aureus* changed the adhesion molecule profile of endothelial cells. Particularly, the expressions of endoglin (ENG) and ICAM-1 remained similar to control, whereas the expressions of VCAM-1 and ELAM significantly grew after 1 h of incubation. ELAM-1 showed the influences of both P- and E-selectins. The TNF results (control) confirmed that the expressions of these receptors were changeable, and therefore, the system worked. The maximum numbers of VCAM-1 and P- and E-selectin molecules were expressed after an hour of endothelial cell incubation with bacteria.

The reason for the decrease in the expression of inducible molecules on endotheliocytes on the second, third, and fourth hours of co-incubation was most likely the appearance of soluble forms of receptors since it has been established that they are in a biologically active form in the blood serum after proteolytic shearing from the cell surface. For example, under bacterial conditioning, the soluble form of the receptors was increased under various inflammatory conditions such as septic shock [26,27].

### 3.2. New System for the Study of Adhesive Contacts between EA.Hy926 Cells and Neutrophils of Healthy Donors Using FS Spectroscopy and Study of the Force and Work of Adhesion

The EA.hy926 cell culture was grown until the achievement of the monolayer (the overgrowth of culture was avoided) to carry out the correct measurements as described in Section 2.2. After its growth, the culture of endothelial cells was washed three times with HBSS under microscopic control. The probe was pretreated with poly-L-lysine, which made it possible to fix the neutrophil granulocyte on the probe. The concentration of 1 × 10^6^ cells/mL neutrophils in the suspension was selected in the series of preliminary experiments because, by using this concentration, only one cell was bound on the probe surface [19]. During the first experiments, the presence of adhesive contacts between the neutrophils and endothelial cells was established (see Section 2.4). The typical experimental curves obtained by using FS spectroscopy are shown in Figure 2.

At a distance of more than 2 μm, a zero value of the force was observed, which indicates the absence of interaction between cells (the cantilever was not curved). Each jump observed on the black curve at a distance of 0.5–2 μm was determined by the rupture of the bonds formed between the neutrophil ligand (the neutrophil attached to the probe) and the endothelial cell receptor (the endotheliocyte attached to the Petri dish surface). All further calculations were performed according to the formulas given in Section 2.4.

After the statistical processing of experimental data, it was found that most of the adhesion force values were less than 100 pN and were determined in the range from 20 to 40 pN. The main part of the adhesion work values was in the range of 130–350 aJ, and none of the values exceeded 600 aJ. A wide range of the force and work of adhesive contacts was a consequence of variability in the donor neutrophils’ adhesive properties because the EA.hy926 endothelial cell culture was a standard culture and could not influence the measured values.

In the work of Fritz et al. (1998), AFM was used for the first time to study the interaction in a group of P-selectin and PSGL-1 receptors. The breaking forces were not constant but continuously increased with the pulling speed up to 165 pN, which was below the calculated maximum force that the P-selectin/PSGL-1 complexes can withstand [28]. However, there were studied pure protein–protein interactions, whereas in our study, the interacting proteins were incorporated into the lipid bilayer membranes of living cells, which significantly changes the physicochemical properties of proteins, including adhesive properties. Therefore, our studies are not completely equivalent. The study of Eibl and Moy (2005) is closer to our work, because the ICAM-1 receptor was functionalized on the surface of the Petri dish, and the LFA-1-expressing cells were attached to the cantilever. The value of the binding forces was about 40 p N in this system, which is comparable to our results [16].

### 3.3. Conditioning Effect of Non-Opsonized S. aureus 2879 M on the Endothelial Cell Expression Profile and on the Adhesion Contacts between Neutrophils and Endothelial Cells

As discussed in Section 3.1, among the ICAM-1, VCAM-1, ELAM-1, and ENG receptors, only the expression levels of VCAM-1 and P- and E-selectin molecules were changed under the *S. aureus* 2879 M condition. The bacterial-induced expression values were comparable to the TNF results, and the maximum expression levels were reached after an hour of incubation. Due to this effect, there was no reason to study the changes in the force and work of adhesion between cells under bacterial conditioning for more than one hour. TNF only induced significantly higher ICAM-1 expression by endothelial cells, and neither bacteria nor TNF could induce the expression of endoglin. Therefore, only VCAM-1 and P-/E-selectins could affect the changes in the force and work of adhesion in the cells’ co-adhesion during the FS spectroscopy experiments under the influence of *S. aureus* 2879 M. However, since VCAM-1 can only bind to the VLA-4 integrin (αDβ2), expressed by monocytes, eosinophils, and lymphocytes, but not neutrophils [29], P- and E-selectins remain the only effectors of adhesive properties in the “neutrophil–epitheliocyte” interaction system. There are selective ligands for these two expressed on the neutrophils’ surface: ESL-1 (a sialylated glycosphingolipid with 5 N-acetyllactosamine repeats (LacNAc, Galβ1-4GlcNAcβ1-3) and 2–3 fucose residues) and sialylated CD44 for E-selectin [30]; CD15 (or sialyl Lewis X) and PSGL-1 (a dimeric mucin-like glycoprotein with a molecular weight of 120 kDa, including a terminal amino acid domain containing an O-glycan with a branched motif containing a sialylated Lewis antigen) for P-selectin [31,32,33]. Thus, changes in the force and work of adhesion between cells under the *S. aureus* condition could be reduced to interactions between P- and E-selectin receptors expressed by endothelial cells and their respective ligands, such as CD15 and PSGL-1, expressed by neutrophils. Using the developed experimental approach, these changes were determined. The results are shown in Figure 3. In view of the individual features of each donor, the changes in both the work and the force of adhesion after incubation with *S. aureus* 2879 M were calculated separately for each experiment. Significant suppression of the force and work of adhesion by bacteria was shown, and in addition, *S. aureus* 2879 M caused a significant decrease in the variability of both the force and the work of adhesion. Table 1 contains the total values of the force and work of adhesion between endothelial cells and neutrophils for all the studied donors either in control or under the *S. aureus* influence. As can be seen, the adhesive contacts weakened between the cells involved in migration through the endothelium.

### 3.4. Conditioning Effect of Opsonized S. aureus 2879 M on the Endothelial Cell Expression Profile and on the Adhesion Contacts between Neutrophils and Endothelial Cells

In [34], it was found that the opsonization of *S. aureus* protects endothelial cells from damage by neutrophils. Therefore, we assumed that the opsonization of *S. aureus* can significantly affect the activity of the bacterium in establishing adhesive contacts between the neutrophil and the endotheliocyte. For these experiments, the bacteria were incubated with pooled sera from healthy donors (at least three to exclude individual variability in the number of opsonins), as described in Section 2.3. As in the case of non-opsonized *S. aureus*, significant variability was observed both in the force and work of the adhesive contacts between the neutrophils and endotheliocytes. The results of the study for individual donors are presented in Figure 4, and the results of calculations of all the force curves from different donors are summarized in Table 2.

The main opsonins (the C3b/iC3b components of complement, IgG (all subclasses), IgM, and IgA [22]) from the serum in this method of strain treatment have their own receptors on the neutrophils, such as CD11b/CD18 and Fcγ (CD64, CD32, and CD16) [35]. Therefore, the preincubation of bacteria with the serum could affect the adhesive contacts between immune cells. However, as shown in Figure 4, although the drop in the force and especially in the work of adhesion was not as critical as that observed under the non-opsonized *S. aureus* condition, the adhesive contacts between neutrophils and endotheliocytes were still significantly reduced after treatment with opsonized *S. aureus*. The mean summary values across all the donors also showed a statistically significant decrease (Table 2).

Thus, both non-opsonized and opsonized *S. aureus* caused the weakening of adhesive contacts between the neutrophils and endotheliocytes, which was not associated with a purely mechanical obstacle, since there were still some differences between the force and work of adhesion after treatment with the non-opsonized and opsonized strains. Thus, we hypothesized that staphylococcus could “cut off” the P- and E-selectins of endotheliocytes, transforming them into soluble forms, but Western blotting showed that the number of soluble forms did not increase (results not shown). Most likely, SSL5 works here, which was described by Bestebroer et al., (2007) [36], or SSL11, as described by Chung et al., (2007) [37]. SSL5 and SSL11 are members of a group of 14 structurally related proteins involved in avoiding the innate immune response. Both of them are able to interact directly with PSGL-1 on both leukocytes and human leukemia HL-60 cells. This interaction cancels the interaction of PSGL-1 with its natural ligand, P-selectin, depending on sulfation and sialylation [36,37,38]. According to the co-crystallization data, it can be concluded that SSL5 or SSL11 are complexed with sialyl Lewis X, which is the key post-translational modification of PSGL-1 binding to P-selectin [39]. *S. aureus* also secretes another molecule, SElX, which interacts with PSGL-1 in a glycosylation-dependent manner and is responsible for avoiding the innate immune response [40]. It has a sialylated glycan-dependent active site homologous to a subfamily in SSLs [41]. This results in neutrophil adhesion through numerous glycosylated surface receptors, leading to impaired IgG-mediated phagocytosis. *S. aureus* also has an extracellular adhesion protein (Eap) that can directly bind to ICAM-1, inhibiting neutrophil recruitment to the site of infection [42].

Moreover, these factors (individually or together) lead to the suppression of adhesive contacts between the neutrophil and the endotheliocyte, which contributes to the partial blockade of a very important stage in transendothelial migration—adhesion. This leads to the arrest of neutrophils in the bloodstream in this experimental septicemia model. Previously, we discovered another factor that significantly reduces the probability of extravasation: a decrease in the stiffness of the membrane–cytoskeleton complex of neutrophils under the condition of the same strain *S. aureus* 2879 M [23]. Thus, one of the options for the pathogenetic strategy of *S. aureus* in the experimental modeling of septicopyemia is to prevent the transendothelial migration of neutrophils from the vascular bed to the extravascular space.

### 3.5. Study of the Viability of Neutrophils after Transendothelial Migration in the Model of Experimental Septicopyemia

Septicopyemia is accompanied by the appearance of purulent foci in tissues and organs. In this case, neutrophils leave the bloodstream and migrate to the site of infection [43]. The migration of neutrophils through a monolayer of the EA.hy926 cell line was investigated in the septicopyemia model, as described in Section 2.6. Additionally, the viability of neutrophils after this migration was estimated. In the control experiment, most of the cells (up to 80%) that migrated to the lower chamber remained viable. In the model of septicopyemia, where *S. aureus* was used as a bacterial chemoattractant, a bacterial concentration of 2.5 × 10^7^ cells/mL was chosen since, for the lower bacterial concentration (2.5 × 10^6^ cells/mL), the migration activity of the neutrophils was similar to that of the control experiment, whereas at a higher bacterial concentration (2.5 × 10^8^ cells/mL), a large amount of cell debris was found.

As can be seen in Figure 5, after the transendothelial migration of neutrophils into a lower compartment of a transwell, these cells often died of classical NETosis (Figure 5b). Additionally, we observed the formation of fast NETs, which was accompanied by the formation of a network-like structure, but the cell remained alive (Figure 5a). As was described in [44], both in fast and classical NETosis, the process proceeds with the active participation of DNA, myeloperoxidase, and citrullinated histones. Therefore, the typical morphology, together with FITC-labeled myeloperoxidase antibody staining, allowed us to conclude that, in the model of septicopyemia induced by *S. aureus*, the neutrophils formed NETs after transendothelial migration. The figure also clearly shows the differences in morphology between fast and classical NETosis: A characteristic cloud-like structure distributed not far from the cell was formed in the case of fast NETosis (Figure 5a), but when the cell died due to the NETosis mechanism, a long network-like structure extending far from the cell appeared (Figure 5b). These results confirmed our previous studies, which showed that the co-incubation of neutrophils with *S. aureus* 2879 M (2 h) caused 14 ± 2% NETosis of the neutrophil population [22].

Thus, passage through the layer of endotheliocytes and subsequent contact with *S. aureus* induced NETosis in neutrophils, and both kinds of NETosis (fast and classical) were observed.

## 4. Conclusions

*S. aureus* 2879 M induced only the expression of the adhesin VCAM-1 and two selectins (P- and E-selectin) on the membranes of the endotheliocytes of the EA.hy926 cell line. The maximum levels of the expression were observed after one hour of co-incubation of endotheliocytes with *S. aureus*. However, an increase in the ligand expression did not compensate for the pathogenetic influence of *S. aureus* on the system, since the adhesive contacts between the neutrophils and endotheliocytes were damaged. In a model of septicemia, when bacteria were in the bloodstream, staphylococci significantly inhibited the interaction between P- and E-selectins on the endothelial cell membrane and PSGL-1, ESL-1, CD15, and CD44, expressed on the membrane of neutrophils. This interaction is a very important step in the chain of reaction of the transendothelial migration process. Therefore, the failure of this stage leads to the impossibility of establishing strong contacts and, as a result, sufficient neutrophil priming for the further process of extravasation. Thus, *S. aureus* is able to induce neutrophil arrest in a model of bacterial septicemia. It is interesting that this effect was somewhat weakened in opsonized staphylococci, although both non-opsonized and opsonized *S. aureus* caused the disruption of adhesion and arrest of neutrophils.

Despite the differences in the force and work of adhesion of neutrophil receptors from the different donors, all the patterns of *S. aureus* influence on cells’ interaction remained the same.

In the model of septicopyemia, the focus of infection was modeled using transwells, and bacteria were used as chemoattractants; when neutrophils passed through endotheliocytes, they often formed NETs. The depth of NETs varied from the survival of cell viability with the formation of fast NETosis to cell death with the formation of classical NETs.

## Figures and Tables

**Figure 1 microorganisms-10-01696-f001:**
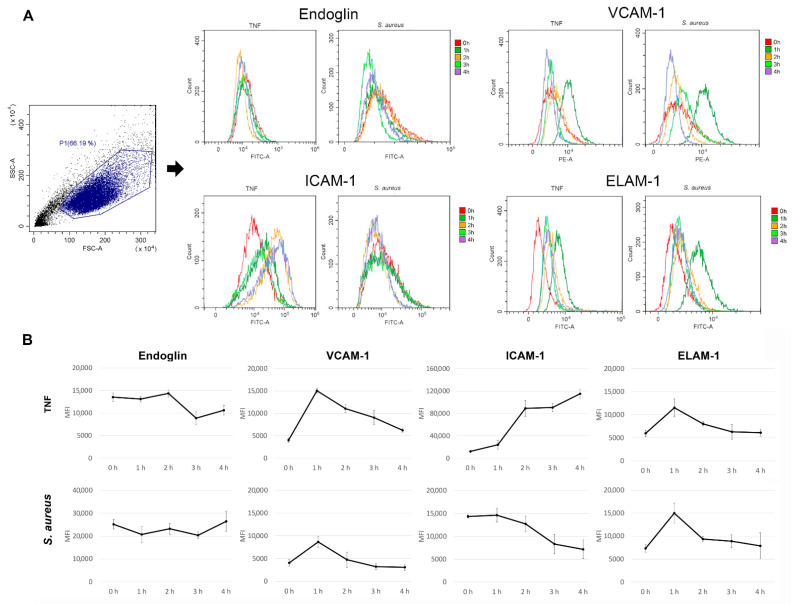
Flow cytometric analysis of adhesion molecules’ expression levels on the endothelial cell surface after incubation with TNF (25 ng/mL) and *S. aureus* 2879 M: (**A**) representative flow cytometry histograms for the endoglin, VCAM-1, ICAM-1, and ELAM-1 expression on the surface of EA.hy926 cells; (**B**) dynamics of endoglin, VCAM-1, ICAM-1 and ELAM-1 expression estimated using median fluorescence intensity (MFI) (error bars represent SD). There were at least 4 experiments for each time point.

**Figure 2 microorganisms-10-01696-f002:**
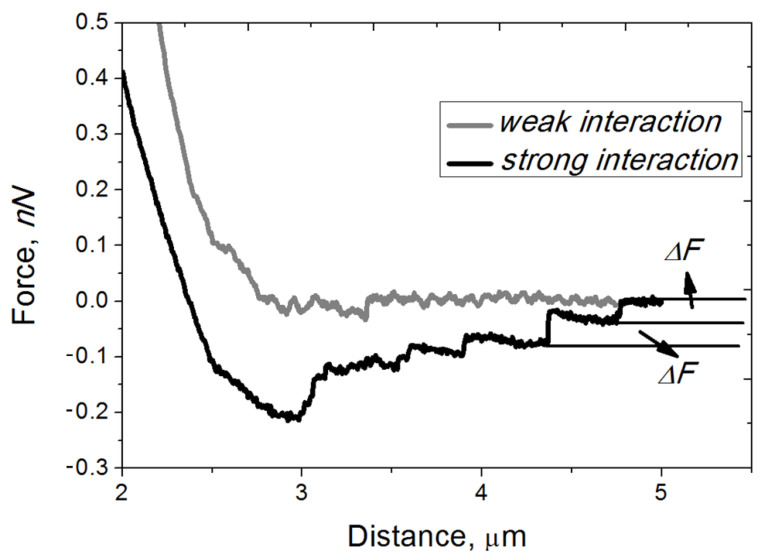
Typical curves depicting the interaction between an endothelial cell and a neutrophil, while the neutrophil was immobilized on the probe and the endotheliocyte adhered to the surface of the Petri dish. The black curve shows a strong adhesive contact between two cells, as steps are clearly visible (Δ*F*). The gray curve shows a weak (non-specific) contact, i.e., there are no steps.

**Figure 3 microorganisms-10-01696-f003:**
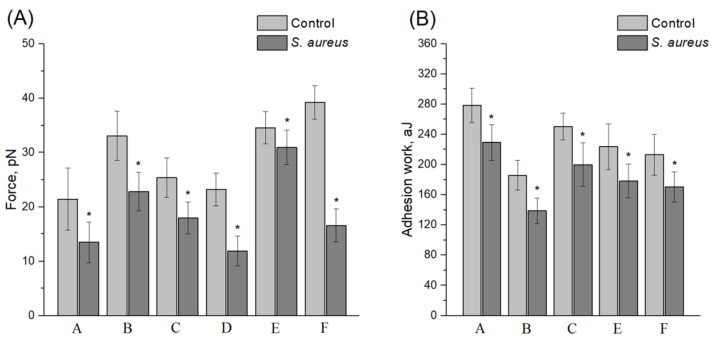
The force (**A**) and work (**B**) of adhesion between the receptors of neutrophils of different donors and ligands of EA.hy926 cell line endotheliocytes show suppression of interaction under non-opsonized *S. aureus* 2879 M. Mean ± SD are shown. A–F show different neutrophil donors. There were at least 5 experiments containing no less than 4 technical replicates for each donor both for control and experiment (minimum of 20 FS curves for each experiment were analyzed). * Significant differences between control and experiment (*p* < 0.05).

**Figure 4 microorganisms-10-01696-f004:**
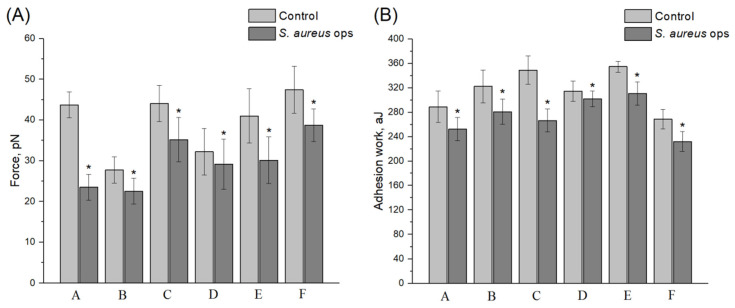
The force (**A**) and work (**B**) of adhesion between the receptors of neutrophils of different donors and ligands of EA.hy926 cell line endotheliocytes show suppression of interaction under opsonized *S. aureus* 2879 M condition. Mean ± SD are shown. A–F show different donors of neutrophils. There were at least 5 experiments containing no less than 4 technical replicates for each donor both control and experiment (minimum of 20 FS curves for each experiment were analyzed). * Significant differences between control and experiment (*p* < 0.05).

**Figure 5 microorganisms-10-01696-f005:**
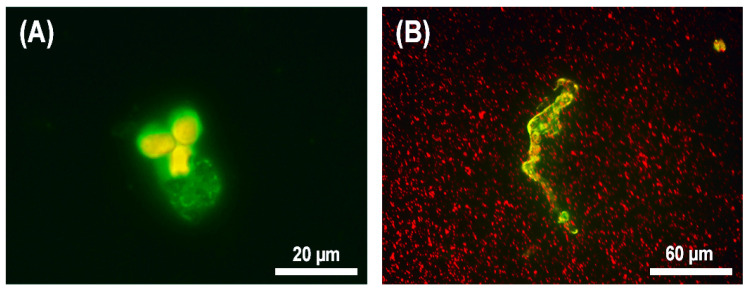
The neutrophil morphology after transendothelial migration in zone with *S. aureus* 2879 M. Neutrophils were stained with propidium iodide and FITC-labeled myeloperoxidase antibodies, and bacteria were used in concentrations of 2.5 × 10^7^ cells/mL (up to 100 bacterial cells per 1 neutrophil in the case of about 10% migration rate): (**A**) Fast NETosis (neutrophil survived); (**B**) classical NETosis (neutrophil died).

**Table 1 microorganisms-10-01696-t001:** The total values of the force and work of adhesion between endothelial cells and neutrophilic granulocytes under the influence of non-opsonized *S. aureus* 2879 M. The table shows the data of 21 force curves for each control and experiment for a series of 5 experiments (cells from different donors were used).

Environment	Force, pN	Adhesion Work, aJ
Vehicle	35.0 ± 11.7	179.6 ± 43.6
*S. aureus*	24.4 ± 6.1 *	134.4 ± 37.8 *

* Significant differences between control and experiment (*p* < 0.05).

**Table 2 microorganisms-10-01696-t002:** The total values of the force and work of adhesion between endothelial cells and neutrophil granulocytes under the influence of opsonized *S. aureus* 2879 M. The table shows the data of 21 force curves for each control and experiment for a series of 6 experiments (cells from different donors were used).

Environment	Force, pN	Adhesion Work, aJ
Vehicle	39.5 ± 7.8	316.3 ± 33.5
*S. aureus*	30.0 ± 6.5 *	274.1 ± 29.8 *

* Significant differences between control and experiment (*p* < 0.05).

## Data Availability

Not applicable.

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
