# Peer review of "Staphylococcus aureus Causes the Arrest of Neutrophils in the Bloodstream in a Septicemia Model"

_microorganisms, 2022, doi:10.3390/microorganisms10091696_

Round 1

Reviewer 1 Report

There are my comments.   The manuscript describes a methodology to study the influence of bacterial pathogens on mammalian cells through the measurement of physical interactions between surface proteins of mammalian cells exposed to bacteria. The methodology is fairly new and has been used in the past. The novelty of the approach is using selected sets of cells and measuring the specific interactions by modified atomic force microscopy. The description of the results is reasonable, and the introduction shows why the experiments were undertaken. The description of results and results suggest that the observed phenomena may have the interpretation suggested by the authors. There are, however, problems with the manuscript that has to be addressed.

  1. The language has been corrected superficially but the punctuation and sentence structure clearly show a need for improvement. It could be easily done by doing a professional review or sending it to a native English speaker.
  2. Figures have been improved. However, Figure 1B is of poor quality and hard to read. It must be improved to at least 300dpi resolution. FACS data are not much better and difficult to tell the difference between different times.
  3. The manuscript outlay and headings are obviously not according to the specifications. It can be corrected during the editorial process.
  4. The word "septicemia" is probably used in the British version of "septicopyemia" and is exchangeable in many places. There is no clear reason why the different names are used or what the supposed difference between them is according to the authors.
  5. The authors suggest that the reason why S. aureus can bypass the normal immune recognition is through the expression of surface molecules preventing normal interaction between different cell types, including those of the integrin family. However, there is known that S. aureus uses a different mechanism bypassing normal peptide recognition through secretion of superantigens. I find it not very credible that the avoidance of immune cell recognition is solely through the modulation of those interactions.
The manuscript is interesting but needs a lot of improvement.   Suggestion: Accept after substantial corrections.

Author Response

Comments and Suggestions for Authors

There are my comments. The manuscript describes a methodology to study the influence of bacterial pathogens on mammalian cells through the measurement of physical interactions between surface proteins of mammalian cells exposed to bacteria. The methodology is fairly new and has been used in the past. The novelty of the approach is using selected sets of cells and measuring the specific interactions by modified atomic force microscopy. The description of the results is reasonable, and the introduction shows why the experiments were undertaken. The description of results and results suggest that the observed phenomena may have the interpretation suggested by the authors. There are, however, problems with the manuscript that has to be addressed.

Dear reviewer, we are very grateful to you for the analysis of our manuscript.

  1. The language has been corrected superficially but the punctuation and sentence structure clearly show a need for improvement. It could be easily done by doing a professional review or sending it to a native English speaker.

Thankyou for the comment, the text has been improved and modified to make it more understandable.

  1. Figures have been improved. However, Figure 1B is of poor quality and hard to read. It must be improved to at least 300dpi resolution. FACS data are not much better and difficult to tell the difference between different times.

The FACS data in Figure 1A is shown as representative and the summary results are shown in 1B to show endoglin, P- and E-selectin and VCAM and ICAM expression over time.The Figure 1 was reworked.

  1. The manuscript outlay and headings are obviously not according to the specifications. It can be corrected during the editorial process.

Thankyou for the comment, the text has been corrected according to the specifications.

  1. The word "septicemia" is probably used in the British version of "septicopyemia" and is exchangeable in many places. There is no clear reason why the different names are used or what the supposed difference between them is according to the authors.

No, here you are absolutely wrong! These are not different versions of English or American, these are two completely different clinical nosological units. Septicemia occurs when bacteria use blood as transport system and signs of intoxication syndrome prevail with severe disorders of microcirculation and central hemodynamics. Septicopyemia – detection of one or more foci of metastatic inflammation with identification of the pathogen. Therefore, in our article, these two clinical pathologies are described as two different models. In the case of modeling septicemia, we added staphylococci between the neutrophil and the endotheliocyte, whereas in the case of septicopyemiamodeling we created a focus (bacteria) "outside the bloodstream".

  1. The authors suggest that the reason why aureus can bypass the normal immune recognition is through the expression of surface molecules preventing normal interaction between different cell types, including those of the integrin family. However, there is known that S. aureus uses a different mechanism bypassing normal peptide recognition through secretion of superantigens. I find it not very credible that the avoidance of immune cell recognition is solely through the modulation of those interactions.

Thankyou. We do not claim that the only mechanism of bypass of normal recognition is the influence of the bacterial cell of S. aureus, but in this work, we studied precisely this process. If we discuss the possibility of your proposal, according to the influence on the bypass of the interaction between the neutrophil and the endothelial cell of superantigens, then it is unlikely, at least in this system. There are 3 classes of superantigens secreted in S. aureus: (1) an exfoliative toxin that destroys desmosomes, but it expresses only by 5% of S. aureus strains and it was not identified in S. aureus 2879 M; (2) toxins causing food poisoning (7 antigen variants: SEA, SEB, SEC1, SEC2, SEC3, SED, SEE), they were found in strains of phage group 3, but S. aureus 2879 M does not belong to it; (3) TSST (toxic shock syndrome toxin) is encoded in the temperate phage; S. aureus 2879 M also does not have it. Thus, in studied system, described in the manuscript, the influence of superantigens is impossible.

The manuscript is interesting but needs a lot of improvement. Suggestion: Accept after substantial corrections.

Reviewer 2 Report

This manuscript describes the use of AFM to study the impact of S. aureus infection on interactions between neutrophils and an endothelial cell line. Overall the results and accompanying methods are described in sufficient detail but a few minor corrections are required.

Figure, 1: Text in section 3.1 indicates significant changes in expression of selected adhesion molecules. Is this statement supported by statistical tests? How many replicates were used for each data point shown in this figure? How many independent experiments were performed?

Section 3.4: “Previously, we discovered another factor that significantly reduces probability of extravasation – a decrease of stiffness of membrano-cytoskeletal complex of neutrophils under condition of the same strain S.aureus 2879 M.” Do you have a reference for this statement? Or is this unpublished work?

Figure 5. Please state a ratio of bacteria to cells (i.e. Multiplicity of Infection), rather than the concentration of bacteria per mL in the figure legend.

Section 4: “However, an increase in ligand expression does not compensate the pathogenetic strategy of S.aureus” Do you mean compensate for? And pathogenic? This section would benefit from some revision. The first  few sentences of the first paragraph are difficult to understand (e.g. the examples just given). Also, the primary focus of the manuscript, as per the abstract and most of the information supplied, is based on AFM data –finishing this section with a summary of the results from Section 3.5 is not the best way to complete this manuscript.

Author Response

Reviewer 2

This manuscript describes the use of AFM to study the impact of S. aureus infection on interactions between neutrophils and an endothelial cell line. Overall the results and accompanying methods are described in sufficient detail but a few minor corrections are required.

Dear Reviewer, thank you for appreciating of our manuscript.

Figure, 1: Text in section 3.1 indicates significant changes in expression of selected adhesion molecules. Is this statement supported by statistical tests? How many replicates were used for each data point shown in this figure? How many independent experiments were performed?

Statistically significant differences were observed for 1-4 hour’s time points for VCAM-1 and ELAM compared to the control, the Mann-Whitney test was used, the experiment was made in 4 replicates.

Section 3.4: “Previously, we discovered another factor that significantly reduces probability of extravasation – a decrease of stiffness of membrano-cytoskeletal complex of neutrophils under condition of the same strain S.aureus 2879 M.” Do you have a reference for this statement? Or is this unpublished work?

Thank you for noting, the reference to this information was added in the text of manuscript.

Figure 5. Please state a ratio of bacteria to cells (i.e. Multiplicity of Infection), rather than the concentration of bacteria per mL in the figure legend.

The ratio of bacteria to cells was added in the Figure 5 capture.

Section 4: “However, an increase in ligand expression does not compensate the pathogenetic strategy of S.aureus” Do you mean compensate for? And pathogenic? This section would benefit from some revision. The first few sentences of the first paragraph are difficult to understand (e.g. the examples just given). Also, the primary focus of the manuscript, as per the abstract and most of the information supplied, is based on AFM data –finishing this section with a summary of the results from Section 3.5 is not the best way to complete this manuscript.

Thankyou for the comment, the beginning of the section has been slightly modified to make it more understandable.We meant that an increase in the expression of ligand molecules, which should enhance the interaction between neutrophils and endotheliocytes, didn’t work, since the affinity of the connection between receptors and ligands of immunocompetent cells was greatly affected by S. aureus. The ending part of the section also was particularly revised.

Round 2

Reviewer 1 Report

The manuscript has been corrected in many parts and s finally readable. The work is exciting, and the experiments are described in a way that can be followed quickly.

I still think the AFM measurements are a combined effect of many factors and are difficult to distinguish from the background. Figure 1 is interpreted later on as saying that there are differences in cell surface expression of surface molecules in the presence and absence of S. aureus. The FACS data in Fig. 1A show it, but the expression dynamics in Fig. 1B are very problematic except for the VCAM-1. I assume that biological data for such systems are challenging to interpret, and small changes are considered significant to fit the authors' interpretation.

There are still many places where the language can be corrected. The journal has an editing service, and this part can be adjusted quickly.

Author Response

Dear reviewer,

Thank you for your comments.

In order to determine the differences between the experimental and control data in both the experiments using AFM and FACS, we used statistical criteria. Statistically significant differences were observed for VCAM-1 and ELAM (P- and E-selectins) at 1 h of incubation with S. aureus compared to the 0 h time point, the Mann-Whitney test was used (P values were 0,029), the experiment was made in 4 replicates. No such differences were found for other molecules. The Student’s test was applied to determine differences in the force and the work of adhesion, at least 20 FS-curves for each experiment were analyzed. Therefore, we consider it legitimate to judge the differences and talk about changes in these values under the influence of S. aureus.

As for English, we gave the article to the editors of a native speaker, he assures us of the correctness of the final text. He asks not just to make a comment about the style of the text, but to say specifically what your comments are about. In any case, we are unable to use the journal's editorial service due to a limited grant that supports our research.

Sincerely yours,

Authors. 

This manuscript is a resubmission of an earlier submission. The following is a list of the peer review reports and author responses from that submission.

Round 1

Reviewer 1 Report

Major comments:

-The organization and the language make the article unreadable. 

 The authors used flow cytometry, however, I could not find the typical dot plots of the flow cytometry analysis. 

-The study also claimed the use of an atomic force microscope, with so much microscopy, there are no good images of the interaction between S. aureus and the endothelial cells. 

Minor comments:   

Sloppy writing. Here are a few examples: 

-line 22: misssing space --> soS.aureus

-Line 76: missing space --> betweenβ2

-Line 78: missing space --> precise,by

-Line 84: missing space --> withessential

-Line 97: missing space --> S.aureusopsonization

-Line 99: cells under S.aureus condition? Or infection?

-Line 106: missing space --> N.Novgorod

-There are several grammatical mistakes and writing issues. 

Reviewer 2 Report

This manuscript describes the use of AFM to examine impact of S. aureus on adherence between neutrophils and epithelial cells, then follows up with some observations on the effects of S. aureus in a transwell system.

The manuscript is difficult to read due to issues with spacing between words; where information has been placed in the manuscript (e.g. the formulas in Section 3.2 would be better placed in the AFM Methods section; some additional explanatory information should be in the introduction); and some of the phrasing. Both the materials and methods and figure legends should provide more information on how many times experiments were repeated and whether data shown are representative or combined from multiple independent experiments.

General comments:

It would be helpful to provide a general introduction to the key adhesins focused on in this manuscript to provide context for the experimental work.

Line 75: MAC-1 and LFA-1 need to be written in full on first usage.

Line 92: what does ‘under S. aureus condition’ mean? Do you mean after infection with S. aureus?

Lines 115-123: is this information specific to the AFM? If so, it should be moved to Section 2.4 or alternatively the purpose of these preparations should be clearly stated.

Line 183: indicate how many S. aureus bacteria were added to the EA.hy926 cell line; and also how many mammalian cells were present.

Sections 2.5 & 2.6: please provide clone names for the antibodies used.

Section 2.5: how were the flow data analysed? Include details of the software used and what information was collected.

Section 2.7 and/or each figure legend: how many times were these experiments repeated?

Figure legends: please indicate that means +/- SD are displayed and confirm whether these are from technical replicates or combined from multiple independent experiments.

Figure 3: This figure presumably shows paired control Vs S. aureus treated samples. If this is the case, the x axis needs to make this clearer – e.g. pairs could be numbered 1 – 6; or A – F. Are these samples the same between Fig 3A and B? Why are there fewer of these in Figure 3B?

Tables 1 & 2: why are these values different to what is shown in Figures 3 and 4? E.g. Fig 3 -Force: Control = 35 Vs S. aureus = 179. Or have these numbers been displayed in the wrong areas of the Table?

Line 406: Note that SSL11 can also bind PSGL-1 - this feature is not unique to SSL5.

Figure 5: The pictures in this figure are a useful illustration of the different outcomes after S. aureus conditioning is included in the trans-well set up, but there is no quantitative data to support the text. How was the extent and/or type of NETosis observed or neutrophil viability quantified? There is not enough information to support the statement in Line 474-476.

Reviewer 3 Report

The manuscript of Pleskova et al. describes the use of atomic force microscopy to study the interaction of neutrophil receptors and ligands of endotheliocytes in the presence of S. aureus. The manuscript has many flaws:

1. Graphics is of poor quality. Fig. 1 is unreadable.

2. Figure 2 does not differentiate between weak and strong interactions. If the data is presented this way, the conclusion is poorly supported.

3. Fig. 3 and 4 have the same legends. What is the difference between the data in Fig. 3 and Fig. 4?

4. Tables 1 and 2 clearly show a difference between controls and the S. aureus for opsonized and non-opsonized cells.

5. Fig. 5 is challenging to interpret. Either the optics is very poor, or the pictures are of poor quality.

6. The premise of the experiments may be summarized in Tables 1 and 2. The rest needs correction (legends) or redesign.

7.  The text has numerous errors and is difficult to follow. It is difficult to say if the writing problems are the results of typographical errors in the manuscript or the assembly process.